# Improving Hepatitis B Screening Prior to Rituximab: A Quality Improvement Project

**DOI:** 10.3390/children10071142

**Published:** 2023-06-30

**Authors:** D. Sofia Villacis-Nunez, Evan Orenstein, Phyllis Selvaggio, Kelly Rouster-Stevens, Chia-shi Wang, Amit Thakral

**Affiliations:** 1Division of Pediatric Rheumatology, Department of Pediatrics, Emory University School of Medicine, Atlanta, GA 30322, USA; diana.sofia.villacis.nunez@emory.edu (D.S.V.-N.);; 2Children’s Healthcare of Atlanta, Atlanta, GA 30329, USA; 3Division of Hospital Medicine, Department of Pediatrics, Emory University School of Medicine, Atlanta, GA 30322, USA; 4Division of Pediatric Nephrology, Department of Pediatrics, Emory University School of Medicine, Atlanta, GA 30322, USA

**Keywords:** hepatitis B screening, rituximab, clinical decision support, clinical informatics, multimodal interventions, quality improvement

## Abstract

Rituximab, used in the treatment of some rheumatic and kidney diseases, can lead to hepatitis B virus (HBV) reactivation; HBV screening is recommended for those starting this medication. We aimed to improve by 50% the proportion of patients undergoing HBV screening by implementing multimodal interventions to support clinicians in this evidence-based practice. We conducted a quality improvement project from November 2020 to June 2022 at a tertiary care pediatric hospital system, including patients with rheumatic and/or kidney diseases starting rituximab. Multimodal interventions targeting clinicians included electronic health tools (dot phrase, display of screening recommendations and screening results in rituximab order sets/therapy plans), educational meetings, and e-mail/paper reminders. The primary outcome was the proportion of patients with complete HBV screening, while the secondary outcome was utilization of each laboratory component, tracked using statistical process control charts. Pre- and post-intervention data were compared using Fisher’s test. One hundred eighty-two patients who had been prescribed rituximab were included, of which 98 (54%) were post-intervention. The proportions of patients undergoing complete HBV screening (6% vs. 44%; *p* < 0.001), HBsAg collection (60% vs. 79%; *p* = 0.006), anti-HBsAb collection (14% vs. 54%; *p* < 0.001), and total anti-HBcAb collection (8% vs. 52%; *p* < 0.001) were significantly higher in the post-intervention period. Improvement was sustained over 18 months, with shifts and/or data points above the control limits in all measures. Forty-five patients were HBV-non-immune. In this study, multimodal interventions including electronic health tools and education of the provider significantly increased the proportion of patients screened for HBV prior to rituximab and identified immunization opportunities.

## 1. Introduction

Immunosuppressive biological medications are frequently used in the treatment of pediatric rheumatic and immune-mediated kidney diseases such as systemic lupus erythematosus with or without lupus nephritis, primary nephrotic syndrome, vasculitis, and others [1,2]. Several infection-prevention strategies are recommended while on biological therapy; among those, screening for chronic and past hepatitis B virus (HBV) infection using hepatitis B surface antigen (HBsAg), anti-hepatitis B surface antibody (anti-HBsAb), and total anti-hepatitis B core antibody (total anti-HBcAb) has been recommended by the Centers for Disease Control and Prevention (CDC) in all patients receiving immunosuppression, due to the potential for HBV reactivation in an immunocompromised state [3]. 

Among the biological drugs, rituximab, a monoclonal antibody against CD20+ B lymphocytes, is considered to convey a higher risk of inducing HBV reactivation as a result of memory B cell depletion and associated increased viral replication [2,4,5]. The presentation of HBV reactivation following rituximab can range from asymptomatic liver function abnormalities to fulminant hepatitis [6]. The highest risk is encountered in HBsAg positive patients. Patients at risk can be offered antiviral therapy [7,8]. Additionally, unvaccinated and non-immune patients (anti-HBsAb negative) are vulnerable to HBV infection and can be identified via HBV serology, which should ideally prompt immunization prior to starting immunosuppressive therapy. 

Pediatric rheumatologists and nephrologists are often responsible for prescribing rituximab as part of the management of various rheumatic and immune-mediated kidney diseases. While data regarding HBV screening practices among these practitioners is lacking, a previous study surveying general rheumatologists suggested that provider awareness and implementation of HBV screening are often suboptimal [9]. Consequently, we designed this quality improvement project to be implemented at our nephrology and rheumatology clinics, aiming to increase the proportion of patients completing HBV screening within our practice.

## 2. Materials and Methods

We conducted a quality improvement project from November 2020 to June 2022, using the Plan-Do-Study-Act (PDSA) methodology. All pediatric patients (age < 21 years) starting rituximab (or its biosimilars) for a rheumatic or kidney disease indication at Children’s Healthcare of Atlanta were included. Children’s Healthcare of Atlanta constitutes the only pediatric tertiary healthcare system and the only nephrology and rheumatology provider serving the Atlanta Metropolitan Statistical Area, the eight largest metropolitan area in the U.S. with approximately 1.5 million people under 18 years of age [10]. The overall goal of the project was to increase the proportion of patients who were screened for HBV prior to receiving rituximab by 50% above the baseline within our population. Pre-intervention data from August 2019 to November 2020 and post-intervention data from December 2020 to June 2022 were tracked. 

Patients prescribed rituximab by a pediatric nephrologist or rheumatologist were identified using electronic medical record automated reports. The reports included rituximab administration setting (e.g., hospital versus infusion center), administration dates, HBV screening orders and results. Participant demographic characteristics and clinical diagnoses were also collected. Reports were manually reviewed by DSVN for accuracy. The primary outcome measure was the proportion of patients receiving rituximab who had a complete HBV screening, defined as the collection of HBsAg, anti-HbsAb, and total anti-HBcAb. Secondary outcomes were the proportions of patients in which each individual laboratory component was utilized. 

### 2.1. Interventions

The interventions were planned by two rheumatology providers (DSVN and AT), who assumed the role of project leaders. Multimodal interventions, both EMR- and non-EMR-based, were implemented between November 2020 and June 2022 (Table 1). As part of the non-EMR- based activities, educational meetings, reminder e-mails, and posted signs as described in Table 1 were designed and implemented by the project leaders; these were aimed at all the pediatric nephrology and rheumatology providers in the practice (faculty and fellows) as well as the nursing staff. 

Considering that rituximab is prescribed at our institution via the use of standardized EMR order groups, known as therapy plans in the outpatient setting and order sets for inpatient, the project leaders then proceeded to design EMR-based clinical decision support tools (Table 1) to assist clinicians in the application of HBV screening guidelines. First, a shared dot phrase detailing the components of complete HBV screening and latest results (if available) was created to assist with ordering and documentation of screening results and was made accessible to all the pediatric nephrology and rheumatology providers. Subsequently, order groups containing the three HBV screening components and an area to display the HBV screening guidelines (Table 1) were outlined and built by the project leaders. In collaboration with EMR analysts, and after undergoing approval by the nursing and physician leadership within the practice, these elements were permanently incorporated into the existing rituximab outpatient therapy plans and inpatient order sets, to be used by all pediatric nephrology and rheumatology providers prescribing rituximab.

### 2.2. Data Analysis

Statistical process control charts plotting the proportion of screened patients over calendar time (*p*-charts) were created to track the proportion of patients with complete HBV screening, as well as the proportion of patients for whom each individual laboratory component was collected; these data were plotted every two months. The average proportion of screened patients in the pre-intervention period was calculated; this was used to establish the central line and to calculate the upper and lower control limits, as described by Mohammed and colleagues [11]. A goal line, marking an increase by 50% above the pre-intervention average was also depicted in each chart. 

These charts were analyzed for the identification of non-random distribution/special cause, indicated by the presence of a shift (eight consecutive data points on one side of the center line), a trend (six or more consecutive data points steadily decreasing or increasing), or a data point outside one of the control limits [12]. Adjustments and recalculations of the center line were to be made if a shift was encountered [13].

Categorical variables are expressed as frequency (percentage) and continuous variables as median (interquartile range (IQR)). Fisher’s exact test were used to compare the proportion of patients with complete HBV screening and collection of individual screening components at baseline versus during the post-intervention period; *p* values ≤ 0.05 were considered statistically significant. 

## 3. Results

Overall, 182 patients who started rituximab for a rheumatic and/or kidney disease indication were included in the analysis (84, 46% in the baseline period and 98, 54%, in the post-intervention period). Demographic and clinical data are detailed in Table 2. Most patients were female (71%) and the median age was 14 years (IQR 10–17 years). The most common diagnosis leading to rituximab initiation was systemic lupus erythematosus with or without lupus nephritis (45%), followed by nephrotic syndrome (32%). Other indications included juvenile dermatomyositis, kidney transplant rejection, antineutrophilic cytoplasmic antibody (ANCA)-associated vasculitis, overlap syndrome, focal segmental glomerulosclerosis, undifferentiated connective tissue disease, polymyositis, atypical hemolytic uremic syndrome, immunoglobulin G4 (IgG4) related disease, panniculitis, pulmonary capillaritis, systemic sclerosis/scleroderma, IgA related vasculitis, polyarticular juvenile idiopathic arthritis, listed in order of frequency. Most infusions were administered in the inpatient setting (83%). 

The proportion of patients with complete HBV screening was significantly higher in the post-intervention period compared to the baseline (44% vs. 6%; *p* < 0.001). Similarly, the proportion of patients who had HBsAg (79% vs. 60%; *p* = 0.006), anti-HBsAb (54% vs. 14%; *p* < 0.001), and total anti-HBcAb (52% vs. 8%; *p* < 0.001) collected prior to rituximab increased significantly in the post-intervention period compared to the baseline. Median time from blood sample collection to results report was one day (IQR 0–1 days). 

The proportion of patients with complete HBV, HBsAg collection, anti-HBsAb collection, and total anti-HBcAb collection over time are presented in Figure 1. Non-random distribution was identified in all charts in the post-intervention period, as noted by several data points above the upper control limit and/or a shift (Figure 1). One or more measurements were at or above the goal in all charts, particularly towards the end of the post-intervention period, roughly corresponding to the introduction of the EMR-based interventions (Figure 1).

No cases of chronic or past/resolved HBV infection were identified. Among patients tested for anti-HBsAb, 8/12 (67%) in the baseline period and 37/53 (70%) in the post-intervention period were found to be non-immune for HBV (anti-HBsAb titers < 10 mIU/mL), for whom an opportunity for immunization was identified. In addition, one patient in the post-intervention period had a positive total anti-HBcAb; her HBV DNA polymerase chain reaction was negative, and her positive result was deemed to be due to chronic intravenous immunoglobulin infusions and not reflective of chronic or past hepatitis B infection.

## 4. Discussion

Vaccination and infectious screening, including testing for chronic or past HBV infection, are integral components of infection prevention in patients with immune-mediated diseases receiving treatment with rituximab; our tertiary care pediatric hospital system provides care for the vast majority of those patients within the large Atlanta area. In this quality improvement project involving an at risk population, we show a significant increase in the proportion of patients screened for chronic/past HBV prior to starting rituximab, through the use of clinical support tools (creation of a dot phrase, display of screening recommendations and screening results in rituximab order sets/therapy plans), and educational interventions followed by reminders to clinicians (e-mail and posted signs in the clinic workspace). To the best of our knowledge, this is the first study to address this important issue in a pediatric population, with a focus on patients with rheumatic and immune-mediated kidney diseases. 

The analysis of baseline HBV screening practices at our hospital system revealed that the proportion of patients with complete screening prior to rituximab was very low (<10%). This gap in care has also been identified among adult rheumatologists and providers from other subspecialties [14,15,16,17]. Component subanalysis was notable for a low proportion of patients whose anti-HBsAb and total anti-HBcAb were measured, but a surprisingly high proportion (>50%) of patients with measured HBsAg. These results are in agreement with a previously reported survey to rheumatology providers showing higher rates of HBsAg use (92%) as part of HBV screening, while the use of anti-HBsAb and total anti-HBcAb were much lower (51 and 43%, respectively) [9]. This is likely due to the lack of awareness among the providers about the guidelines for adequate screening for chronic and past HBV infection. 

At our institution, order panel availability may also have played a role in the comparatively higher proportion of HBsAg screened patients versus the other two screening components. Locally, most providers utilize a panel that measures acute serologic markers for hepatitis A, B, and C as screening prior to rituximab; this panel only captures HBsAg and anti-HBcAb IgM, which do not constitute adequate screening for chronic or past infection. Through our interventions, we were able to increase awareness about this common pitfall, which led to increase in utilization of all screening components. Nonetheless, post-intervention use of HBsAg collection remained more common compared to the rest of the screening components; EMR-embedded tools should help address this disparity over time. 

Interestingly, as noted in the statistical process control charts, the initial non-EMR based interventions had little effect on our outcomes in the early post-intervention period. This is in contrast to a previous report of adult patients receiving rituximab showing that guideline creation and provider education alone led to a significant increase in HBV screening rates; however, only two post-policy introduction measurements are reported in that study, limiting the potential to analyze the effect of their interventions over time [18]. Non-random increments in the proportion of screened patients (with measurements >50% above baseline) were identified in the late post-intervention period, suggesting efficacy of our interventions, particularly those that constituted clinical decision support tools and were EMR-based. The displayed screening guidelines and readily available screening orders/results within the rituximab therapy plans and order sets likely acted as timely, constant reminders to the providers, thus leading to the increments in the number of screened patients as noted in our results. These EMR-based tools will remain permanently embedded in the EMR and are anticipated to generate a sustained improvement of the HBV screening practices within our hospital system.

Noticeably, no cases of hepatitis B were identified in our cohort. This is consistent with previously reported low prevalence of HBV infection in pediatric rituximab recipients in United States studies (1 case out 2875 patients in a 12-year period) [19]. Despite this low prevalence, given the high risk and potentially severe complications from HBV reactivation, universal screening prior to rituximab remains crucial. Furthermore, we identified several patients in need of HBV immunization who would have otherwise remained unrecognized. This further highlights the importance of serologic testing for HBV prior to therapy with rituximab. 

Future study designs addressing this topic should aim to expand this initiative across other subspecialties within our healthcare system to implement a similar process and ensure an adequate screening system, which would allow to assess the efficacy of our interventions in an even larger population. Moreover, the inclusion in this project of other centers with randomization of the interventions would allow to further strengthen our model with potential for a nationwide impact. 

### Limitations

First, this study was conducted in a single healthcare system, which may have limited the generalizability of our findings. However, considering that Children’s Healthcare of Atlanta is the only pediatric nephrology and rheumatology provider for the large Atlanta area, our patient sample is substantial, and a similar number of patients would be unlikely to be observed in smaller practices. 

Additional factors that could restrict the applicability of our project to other institutions include the possible differences of EMR software and functionality unique to each center and practice, which should be considered at other sites looking to implement this project. Nonetheless, essential lessons learned through our project, including the involvement of key stakeholders, the use of institutional bioinformatics support, and identification of an EMR component with potential to reach all ordering providers, are strategies that could be used to successfully launch this project at other sites. 

It is also worth mentioning that our multimodal design, with several interventions introduced throughout the project, limits our ability to determine whether the EMR-based interventions alone would have been beneficial without being preceded by or combined with provider education and non-EMR based interventions. Nonetheless, published literature supports the use of multifaceted interventions to influence change in providers’ behavior; single interventions, particularly those that are passive and involve education alone not followed by reminders, have been shown to be less likely to succeed [20], which supports the use of our multimodal design in future projects.

Lastly, though many of our interventions were EMR-based, they remained provider-dependent, which can still lead to variability in screening practices and missed screening and immunization opportunities. Provider-independent or semi-independent EMR-based tools able to detect missing results and automatically place screening orders have led to an increase in screening rates across multiple subspecialties in a previous report [21]. These tools could be considered in the future at our institution to build on our screening model. 

## 5. Conclusions

Screening for chronic and past HBV infection is a critical step in the evaluation of patients receiving rituximab. In this study, clinical decision support tools combined with educational interventions significantly increased the proportion of patients screened for chronic and past HBV infection prior to receiving rituximab for a rheumatic and/or immune-mediated kidney disease. In addition, we identified opportunities for immunization in >50% of the patients undergoing screening, which would have been overlooked in the absence of serologic testing. Permanent EMR-based tool changes are anticipated to drive a sustained increase of HBV screening rates long term within our large, tertiary care practice. 

## Figures and Tables

**Figure 1 children-10-01142-f001:**
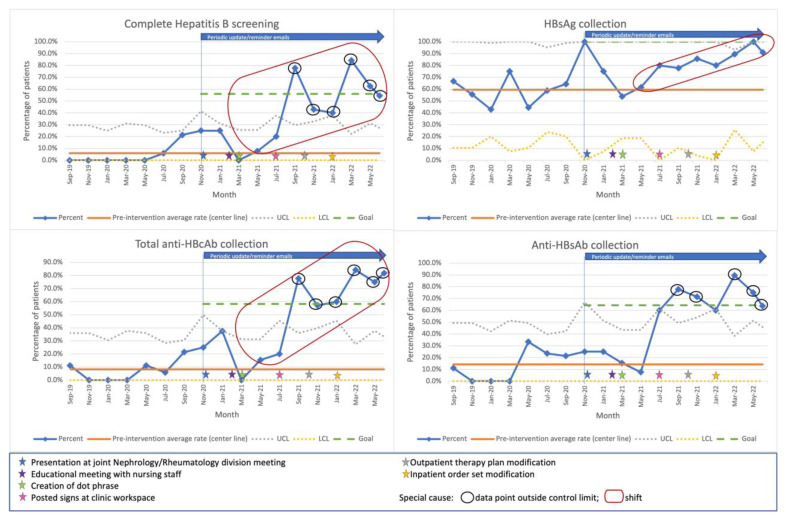
Statistical process control charts plotting the proportion of patients screened for chronic or past Hepatitis B virus infection and use of individual screening components over calendar time. Interventions are marked by star shapes and described in the bottom panel. Non-random distributions were identified as data points above the upper control limit and/or a shift in all charts. Center line recalculation after a shift is not depicted as it corresponds to the post-project completion period. Upper control limit shown as 100 if ≥100. Lower control limit shown as 0 if ≤0. Abbreviations: anti-HBcAb: hepatitis B anti-core antibody; anti-HBsAb: Hepatitis B anti-surface antibody; HBsAg: Hepatitis B surface antigen; LCL: Lower control limit; UCL: Upper control limit.

**Table 1 children-10-01142-t001:** Description of the implemented multimodal interventions.

Intervention	Date	Description
Non-electronic medical-record-based		
Project presentation	November 2020	Meeting with the pediatric nephrologist and rheumatologists describing the project and promoting participation
Educational meeting	February 2021	Meeting with the nursing staff describing the project and promoting participation
Posted signs	July 2021	Printed signs detailing the recommendations for universal HBV screening prior to rituximab and its components, along with the contact information for the project leads were posted at the clinic workspace
Periodic e-mails	November 2020 to June 2022	E-mails containing reminders about the recommendations for universal HBV screening prior to rituximab and its components were sent to the pediatric nephrologists and rheumatologists in the post-intervention period
Electronic medical-record-based: Clinical decision support tools		
Dot phrase	March 2021	Created to include the following text: “Hepatitis B screening prior to Rituximab therapy: Screening for chronic or past hepatitis B using HBsAg, anti-HBc total Ab, AND anti-HBs Ab should be performed up to 8 weeks prior to starting Rituximab therapy. This patient [has/has not] been screened for chronic/past hepatitis B infections within the last 8 weeks. Accordingly, pertinent screening orders [were/were not] placed during this visit. [Chronic/past hepatitis B screening results for this patient are as follows: ***]”
Modifications to rituximab outpatient therapy plans (orders added)	October 2021	Nursing communication order: “For Rheumatology and Nephrology patients, FIRST INFUSION ONLY: Please verify that Hepatitis B Virus Surface Ag screen, Hepatitis B Core Total Ab, AND Hepatitis B Surface Ab have been screened within the last 8 weeks. Otherwise, please collect missing labs prior to infusion.” Orderable screening laboratories under the laboratory section: Hepatitis B Virus Surface Antigen screen, Hepatitis B Core Total Antibody, Hepatitis B Surface Antibody
Modifications to rituximab inpatient order sets (orders added)	January 2022	Subsection header within the “Labs” section: “Hepatitis B Screening Labs Prior to first Rituximab infusion only: patient should have Hepatitis B surface antigen (HBsAg), anti-Hepatitis B surface antibody (anti-HBs Ab), AND total anti-Hepatitis B core antibody (total anti-HBc Ab) tests checked within the last 8 weeks. If one or more of these labs are missing, please order before starting the infusion. Last results (if available) are shown below: ***” Orderable screening laboratories under the “Hepatitis B Screening Labs subsection”: Hepatitis B Virus Surface Antigen screen, Hepatitis B Core Total Antibody, Hepatitis B Surface Antibody

***: If available, this area would display the latest screening results.

**Table 2 children-10-01142-t002:** Demographics, clinical characteristics, HBV screening orders and results.

Category	Total (*n* = 182)	Pre-Intervention (*n* = 84)	Post-Intervention (*n* = 98)
Demographics			
Female	115 (71)	50 (60)	65 (66)
Age (years)	14 (10–17)	14 (10–16)	14 (11.3–17)
Diagnosis			
Systemic lupus erythematosus with or without lupus nephritis	82 (45)	41 (49)	41 (52)
Nephrotic syndrome	48 (29)	27 (32)	26 (26)
Other	40 (22)	16 (19)	24 (24)
Setting			
Inpatient	135 (83)	65 (77)	70 (71)
Outpatient infusion center	47 (29)	19 (23)	28 (29)
HBV screening orders			
Complete HBV screening	48 (29)	5 (6)	43 (44)
HBsAg collected	127 (78)	50 (60)	77 (79)
Anti-HBsAb collected	65 (40)	12 (14)	53 (54)
Total anti-HBcAb collected	58 (36)	7 (8)	51 (52)
HBV screening results (among patients undergoing serologic testing)			
HBsAg non-reactive	127 (100)	50 (100)	77 (100)
Anti-HBsAb < 10 mIU/mL	45 (69)	8 (67)	37 (70)
Anti-HBsAb > 10 mIU/mL	18 * (28)	4 * (33)	14 * (26)
Anti-HBsAb reactive	2 (3)	0 (0)	2 (4)
Total anti-HBcAb non-reactive	57 (98)	7 (100)	50 (98)
Total anti-HBcAb reactive	1 (2)	0 (0)	1 (2)

* Patients with prior HBV immunization history. Categorical variables are expressed as *n* (percentage) and continuous variables as median (IQR). Abbreviations: anti-HBcAb: hepatitis B anti-core antibody; anti-HBsAb: Hepatitis B anti-surface antibody; HBsAg: Hepatitis B surface antigen; IQR: Interquartile range.

## Data Availability

The data presented in this study are available on request from the corresponding author.

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
