# Peer review of "Improving Hepatitis B Screening Prior to Rituximab: A Quality Improvement Project"

_children, 2023, doi:10.3390/children10071142_

Round 1
Reviewer 1 Report
Considering that Rituximab is among the most frequently used immunotherapy in pediatric population and the fact that further strategies to prevent the risk of infection following this type of treatment should be developed, your study could contribute to this direction. The aim of the research, the methodology and results were clearly described, and the conclusions were well supported by your findings. Only one remark, please explain the meaning of the category "other" from Table 2, especially because it is a clear increase of its value post-intervention.
Author Response
Thank you for your comments, we appreciate your time. A detailed description of the conditions included in the category Other in Table 2 has been listed in the body of the manuscript – results section.
Reviewer 2 Report
Thank you for the opportunity to review this well written manuscript describing a well designed QA project designed to increase HBV screening prior to initiation of rituximab in at risk children with rheumatic disease or renal dysfunction. Your introduction clearly establishes the importance of the project, and your references are pertinent and current. The study design, methods, and timeline are described effectively, along with an outstanding explanation of statistical methods and data analysis. Your results are clear and concise, and the control charts are especially helpful in conveying the key findings of your study. The discussion and limitations sections are very helpful, and I agree with your conclusions. This was a pleasure to read, and I have no changes to recommend.
Author Response
Thank you for your comments, we appreciate your time.
Reviewer 3 Report
Dear authors,
it is certainly an interesting and current topic. As a pediatric nephrologist, I used rituximab and consequently tested children for latent HBV infection. I consider the subject extremely important in the case of immunosuppressive therapy.
Returning to the proposed article:
- the summary respected the required typology and the recommended number of words
- the introduction could be extended
- material and method - describes the way of working and the methods applied accordingly
- the results are rendered synthetically
- the discussions could be extended by including new references.
A quick search on Pubmed turned up many articles on this topic. International guidelines recommend HBV screening before rituximab treatment and subsequent antiviral prophylaxis among patients with a (resolved) infection.
Unfortunately, many patients treated with rituximab are not properly screened for HBV infection and often do not receive antiviral prophylaxis. Although screening rates have improved over time, rates remain suboptimal in the literature, even in countries where the incidence is low. With the increasing number of indications for rituximab and other immunosuppressive agents, these findings may increase awareness among all medical specialties prescribing these agents. I congratulate you on the article. With a minimal effort in improving the discussion section and, implicitly, the references.
Author Response
Thank you for your comments, we appreciate your time. We have updated the introduction and discussion to include your remarks and have added some pertinent references to the manuscript.